



# γ-effects identify preferentially populated rotamers of CH₂F groups: side-chain conformations of fluorinated valine analogues in a protein

Elwy H. Abdelkader[1], Nicholas F. Chilton[2,3], Ansis Maleckis[4], Gottfried Otting[1]

[1] ARC Centre of Excellence for Innovations in Peptide & Protein Science, Research School of Chemistry, Australian National University, Canberra, ACT 2601, Australia

[2] Research School of Chemistry, Australian National University, Canberra, ACT 2601, Australia

[3] Department of Chemistry, The University of Manchester, Manchester M13 9PL, United Kingdom

[4] Latvian Institute of Organic Synthesis, Aizkraukles 21, LV-1006 Riga, Latvia

*Correspondence to*: Gottfried Otting (gottfried.otting@anu.edu.au)

**Abstract.** Using cell-free protein synthesis, the protein G B1-domain (GB1) was prepared with uniform high-level substitution of valine by (2*S*,3*S*)-4-fluorovaline, (2*S*,3*R*)-4-fluorovaline, or 4,4'-difluorovaline. The $^{19}$F nuclear magnetic resonance (NMR) signals are distributed over a wide spectral range. The fluorinated samples maintain the relative $^1$H chemical shifts of the wild-type protein, opening a convenient route to assigning the $^{19}$F NMR signals. For the singly fluorinated residues, the $^{13}$C chemical shifts of the remaining CH₃ group are subject to a γ-effect that depends on the population of different rotameric states of the CH₂F group and correlates with $^3J_{FC}$ coupling constants. In addition, the preferentially populated rotamers are reflected by the γ-gauche effect on $^{19}$F chemical shifts, which correlates with $^3J_{HF}$ couplings. Some of the side-chain conformations determined by these restraints position the fluorine atom near a backbone carbonyl group, a non-intuitive finding that has previously been observed in the high-resolution crystal structure of a different protein. Through-space scalar $^{19}$F–$^{19}$F couplings due to transient fluorine–fluorine contacts are observed between residues 39 and 54.

## 1 Introduction

Fluorine atoms incorporated into proteins provide convenient probes for monitoring and analysis by $^{19}$F-NMR spectroscopy. As CF and CH groups in organic compounds feature notably similar spatial requirements and hydrophobicities, the global substitution of a single amino acid type in a protein by a selectively fluorinated analogue is usually possible with minimal structural perturbation and limited penalty in fold stability. If the fluorine atom is installed in a methyl group, the resulting CH₂F group has the freedom to respond to the increased spatial requirement of the fluorine atom by preferential population of those rotamers that are most readily accommodated by the chemical environment. Recently, we showed that the 19 kDa protein *E. coli* peptidyl-prolyl isomerase B (PpiB), which contains five leucine residues, can be produced with high-level uniform substitution of leucine for (2*S*,4*S*)-5-fluoroleucine (FLeu1), (2*S*,4*R*)-5-fluoroleucine (FLeu2) or 5,5'-difluoroleucine (diFLeu;





Fig. 1) by using cell-free protein synthesis (Tan et al., 2024). As demonstrated by X-ray crystal structures, the structural perturbations caused by these amino acid substitutions were minimal (Frkic et al., 2024a). The $^{19}$F-NMR signals were dispersed over a large chemical shift range and acted as highly sensitive probes of ligand binding (Tan et al., 2024). Similarly, when PpiB was produced with (2*S*,3*S*)-4-fluorovaline (Fig. 1), the melting temperature of the protein decreased by no more than 15 °C despite the substitution of sixteen valine residues and the crystal structure revealed a fully conserved protein fold. The

spectral range of the $^{19}$F-NMR spectrum exceeded 20 ppm (Frkic et al., 2024b).

Among the light elements with spin 1/2, fluorine is unusual in that contacts between two $^{19}$F spins readily produce through-space scalar $^{19}$F–$^{19}$F couplings, $^{TS}J_{FF}$ (Hierso 2014), which can also be observed in proteins (Kimber et al., 1978; Orton et al., 2021). In GB1 made with fluorinated leucine analogues in place of canonical leucine, $^{TS}J_{FF}$ couplings between $C^\delta H_2F$ groups are readily manifested in [$^{19}$F,$^{19}$F]-TOCSY spectra although the couplings are small (up to about 3 Hz; Tan et al., 2025).

In agreement with a significant energy barrier between staggered rotamers calculated for 1-fluoropropane (Feeney et al., 1996), the crystal structures of PpiB made with fluorinated leucine and valine residues show that the $CH_2F$ groups strongly prefer one of the three staggered rotamer conformations while frequently populating more than single rotamers. As $^{TS}J_{FF}$ couplings depend on short-range orbital overlap (Hierso 2014), the small size of $^{TS}J_{FF}$ couplings may signal transient fluorine–fluorine contacts between non-uniformly rotating $CH_2F$ groups.

To gain a better understanding of the preferential populations of different rotamer conformations of $CH_2F$ groups under solution conditions at room temperature, we produced GB1 with fluorinated valine residues. Figure 1 shows the fluorinated valine analogues used, which were installed by producing GB1 by cell-free synthesis from a mixture of the twenty amino acids with canonical valine substituted by one of the fluorinated analogues. The results show that (i) the γ-effect of $^{13}$C chemical shifts (Fürst et al., 1990; Günther 2013) provides a readily accessible restraint to define the preferential rotamers

populated by a $CH_2F$ group, (ii) multiple rotamers are usually populated, (iii) there appears to be a bias towards rotamers positioning the fluorine atom near the positively polarized carbon of a carbonyl group and (iv) $^{TS}J_{FF}$ couplings can be observed between neighbouring fluorovaline residues. The results highlight the potential of fluorinated amino acids as NMR probes that minimally perturb protein structure.





(2S,3S)-4-fluorovaline    (2S,3R)-4-fluorovaline

4,4'-difluoro-L-valine


**Figure 1.** Chemical structures of the fluorinated valine analogues used in the present work. (2S,3S)-4-fluorovaline, (2S,3R)-4-fluorovaline and 4,4'-difluoro-L-valine are referred to in the following as FVal1, FVal2 and diFVal, respectively. Samples of GB1 made with uniform substitution of valines for FVal1, FVal2 and diFVal are referred to as GB1-1, GB1-2 and GB1-d, respectively.


## 2 Results

### 2.1 Protein production, stability and NMR samples

The preparation of the proteins GB1-1, GB1-2 and GB1-d and mass spectrometric analysis has been described previously (Maleckis et al., 2022). Subsequent preparations used commercial FVal1 and FVal2 from Enamine (Ukraine). The respective proteins were made by cell-free protein synthesis replacing valine by either FVal1, FVal2 or diFVal at 2 mM concentration. Mass-spectrometry analysis of the intact proteins showed that the most abundant species was the protein where every valine was substituted by the fluorinated analogue. A significant fraction featured one canonical valine residue among the four valine

sites in the protein, amounting to about 30% of the main species in GB1-1 and GB1-2, but 80% in the case of GB1-d, indicating that increasing fluorination challenges the recognition of the amino acids by the *E. coli* valyl-tRNA synthetase. In the case of GB1-d, random installation of one canonical valine together with three diFVal residues would thus yield four different species, each about five-fold less abundant than the main GB1-d product where all valine residues are replaced by diFVal.

Monitoring the heat denaturation of the proteins by circular dichroism indicated melting temperatures $T_m$ between 67

ºC and 70 ºC, i.e., about 10 ºC lower than for the wild-type protein under the same conditions (Fig. S1; Tan et al., 2024).



The NMR samples were prepared in 90% $H_2O$/10% $D_2O$, 100 mM NaCl, 20 mM MES buffer pH 6.5 and 0.1 mM trifluoroacetate as calibration standard (-75.25 ppm). The protein concentration of the final NMR samples was about 0.7 mM (1.6 mM for HOESY spectra of GB1-1 and GB1-2).

## 80   2.2 1D [19]F-NMR

The 1D [19]F-NMR resonances of GB1-1 and GB1-2 are distributed over a large chemical shift range and well resolved, except for a single instance of signal overlap in GB1-1, and all eight expected [19]F-NMR signals are resolved in GB1-d (Fig. 2). The greater chemical heterogeneity of the GB1-d sample is evidenced by additional peaks of about 20% intensity (Fig. 2c). In previous work on GB1 produced with diFLeu, the [19]F chemical shifts were found to strongly depend on whether nearby leucine
sites in the protein contain canonical leucine or the fluorinated analogue (Tan et al., 2025). Therefore, the low intensity peaks in the [19]F-NMR spectrum of GB1-d most likely originate from the species with one valine and three diFVal residues.

[19]F-NMR spectra recorded without [1]H decoupling display broad multiplets. Fundamentally, each $C^\gamma H_2 F$ group produces a triplet due to splittings by the $^2J_{HF}$ coupling constant (about 47 Hz) which is split further by the $^3J_{HF}$ coupling with the $H^\beta$ atom. If the $^3J_{HF}$ coupling is of comparable magnitude as the $^2J_{HF}$ coupling, the multiplet appears like a quartet. An
example is the [19]F-NMR signal of the $\gamma_2$-fluorine of residue 54 (Fig. 3b and c). In contrast, the $\gamma_1$-fluorine of this residue appears like a triplet of doublets, indicating a much smaller $^3J_{HF}$ coupling constant (Fig. 3a). The inverse correlation of $^3J_{HF}$ couplings with [19]F chemical shifts is a manifestation of the γ-gauche effect predicted by Oldfield and co-workers based on quantum calculations (Feeney et al., 1996) and experimentally confirmed for fluorinated leucine residues in the proteins PpiB (Tan et al.; 2024; Frkic et al., 2024a) and GB1 (Tan et al., 2025), and for FVal1 in PpiB (Frkic et al., 2024b). In the case of GB1 with
fluorinated valine residues, the correlation between 3-bond couplings and [19]F chemical shifts is less striking. For example, residue 29 in GB1-2 produces the most high-field shifted resonance, but the multiplet indicates a smaller 3-bond coupling than for residue 54 (Fig. 4b).

Inversion-recovery experiments of GB1-d indicated $T_1(^{19}F)$ relaxation times ranging between 0.35 s (residue 39) and 0.6 s (residue 54), i.e. slightly slower than for GB1 with difluoroleucine, where $T_1(^{19}F)$ was about 0.3 s (Tan et al., 2025).
$R_{1\rho}(^{19}F)$ measurements of GB1-d ranged between 9 s$^{-1}$ ($\gamma_2$-fluorine of residue 21) and 26 s$^{-1}$ ($\gamma_2$-fluorine of residue 54; Table S2). On average, these rates are faster than those of diFLeu residues previously studied in the same protein (Tan et al., 2025), which may be attributed to the greater proximity of the $CH_2F$ groups of diFVal to the protein backbone, thus limiting side-chain mobility. The broadest signals were observed for the most deeply buried residues, indicating that the peak heights are sensitive indicators of the side-chain mobilities (Fig. 2).






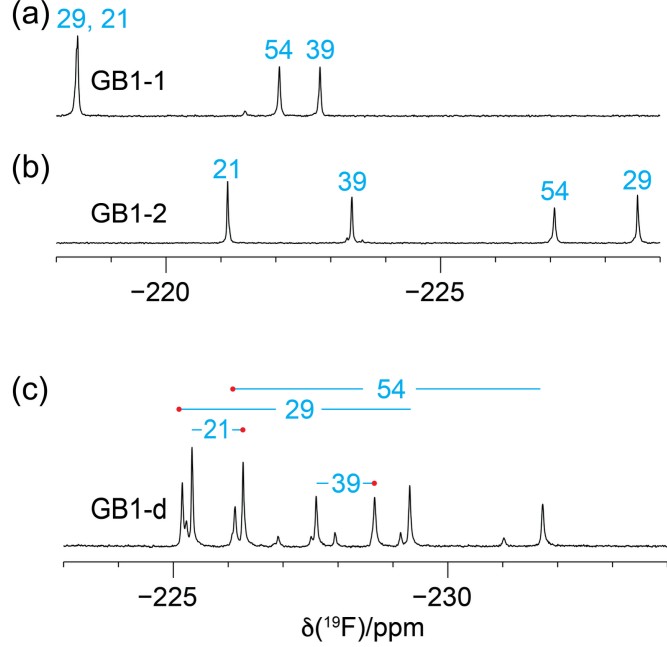

**Figure 2.** 1D $^{19}$F-NMR spectra of GB1 made with fluorinated valine analogues. Unless stated otherwise, all NMR spectra in this work were recorded at 25 °C on a 400 MHz NMR spectrometer with $^1$H-decoupling. The peaks are labelled with the sequence-specific resonance assignments (blue). (a) GB1 made with FVal1. (b) GB1 made with FVal2. (c) GB1 made with diFVal. Red dots identify the resonances assigned to the C$^{\gamma1}$H$_2$F groups.



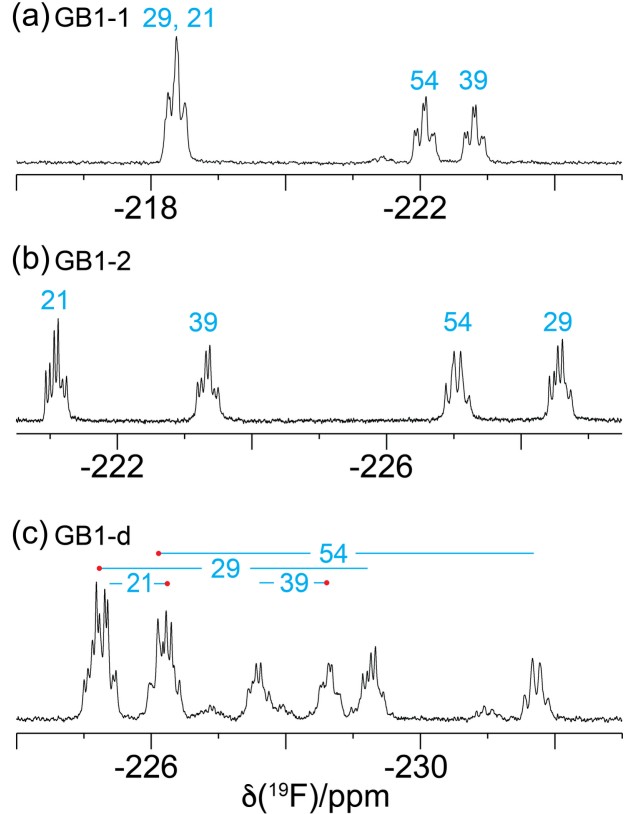

**Figure 3.** 1D $^{19}$F-NMR spectra recorded without $^1$H decoupling. Same samples and conditions as in Fig. 2.


### 2.3 Resonance assignments

The sequence-specific resonance assignment of the $^1$H-NMR spectra of GB1-1, GB1-2 and GB1-d were readily obtained by conventional 2D NMR spectroscopy, using NOESY (200 ms mixing time), TOCSY (80 ms mixing time) and DQF-COSY experiments recorded on an 800 MHz NMR spectrometer. Structural conservation was indicated by conserved chemical shifts.

For example, ring currents induced by W43 cause characteristic high-field shifts of the H$^\beta$ and C$^{\gamma2}$H$_3$ resonances of V54 (to about -0.3 ppm and 0.4 ppm, respectively) and these high-field shifts persist in the samples made with fluorinated valine analogues (Fig. S2). Specifically, all fluorinated valine analogues appear to maintain the side-chain conformations of the wild-type protein as indicated by the intensities of their H$^\alpha$–H$^\beta$ COSY cross-peaks, which were intense for residues 29 and 39 (indicative of a dihedral angle $\chi_1$ of about 180°), but weak or below the level of detection for residues 21 and 54 (as expected

for a small $^3J$(H$^\alpha$,H$^\beta$) coupling) in accordance with the conformations shown in Fig. 4b.





The $^{19}$F-NMR signals were readily linked to the assigned $^{1}$H-NMR spectra using short-delay $^{1}$H,$^{19}$F correlation experiments (Fig. 5; Tan et al., 2024) and [$^{1}$H,$^{1}$H]-TOCSY spectra followed by an INEPT transfer to $^{19}$F (Fig. S2). These experiments were particularly important for GB1-d as they provided intra-residue links between $^{19}$F resonances (Fig. 5c).

As expected for the electron withdrawing effect of fluorine, the $^{1}$H-NMR signals of the fluorinated residues were
shifted low-field relative to the wild-type protein. The $^{1}$H$^{\beta}$ resonances of the singly fluorinated residues were shifted by about 0.3 ppm and those of the diFVal residues by about 0.5 ppm, except for residue 54, which is exposed to ring currents from W45. The $^{1}$H$^{\alpha}$ signals were shifted low-field by about 0.25 ppm and 0.45 ppm in the singly and doubly fluorinated residues, respectively.

The relative chemical shifts of the protons of the CH$_2$F groups were highly conserved in all three samples (Fig. 5a–c),
with their average chemical shifts increasing in the following order of residues: 54 < 39 < 29 < 21, the only exception being the $\gamma_1$-protons of residue 54 in GB1-d, which were slightly low-field of the $\gamma_2$-protons of residue 39 (Fig. 5c). The ordering of $^{1}$H chemical shifts is also conserved relative to the wild-type protein, if the average of the chemical shifts of both methyl groups in each valine residue of the wild-type protein is used as the reference. Likewise, the chemical shifts of the H$^{\beta}$ atoms follow the ordering in the wild-type protein. This highlights the conservation of the amino acid side chain conformations in
the 3D structure of the protein. Within a CH$_2$F group, the $^{1}$H chemical shifts were often different from each other (Fig. S3).

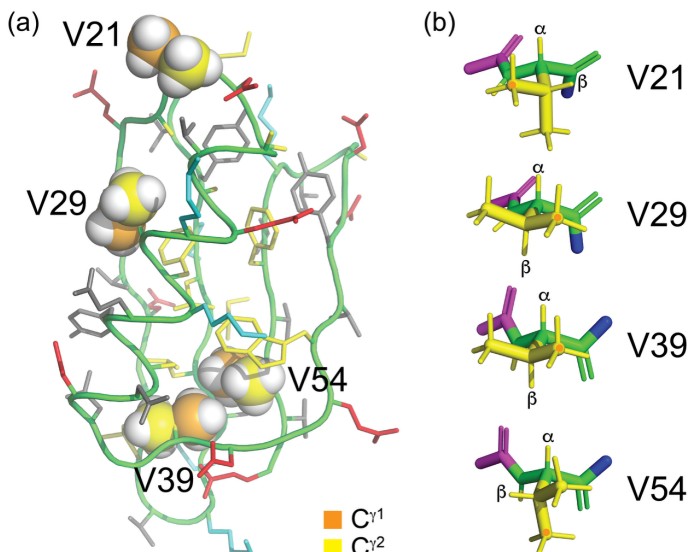

**Figure 4.** Solution structure of GB1 (PDB ID 3GB1, Juszewski et al., 1999). (a) Overview showing the methyl groups of the four valine residues in a space-filling representation. The $\gamma_1$- and $\gamma_2$-carbon atoms are coloured orange and yellow, respectively.
V39 and V54 are buried, whereas V29 is partially solvent-exposed and V21 is highly solvent-exposed. The side chains of the other amino acids are coloured blue (positively charged), red (negatively charged), grey (hydrophilic) and yellow (hydrophobic). (b) Stick representation of the four valine residues, showing the backbone N, C$^{\alpha}$, C' and O atoms in green and



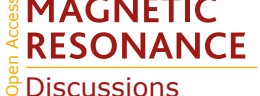

the side chain in yellow. The nitrogen of the previous residue is shown in blue and the $C^\alpha$, C' and O atoms of the following residue in magenta. The rotamer states of the methyl groups, which vary between different conformers in the solution structure,

were rotated to depict the ideal staggered conformations. The $H^\alpha$ and $H^\beta$ atoms are labelled and the $C^{\gamma 1}H_3$ groups are marked with an orange dot. In three of the valine residues, the valine carbonyl carbon is close to a methyl hydrogen, involving the $C^{\gamma 1}H_3$ group of residues 29 and 39 and the $C^{\gamma 2}H_3$ group of residue 54.

**Figure 5.** Short-delay $^1$H,$^{19}$F correlation experiments (Tan et al., 2024) for measuring $^3J_{HF}$ couplings and $^{19}$F-detected [$^1$H,$^1$H]-TOCSY spectrum for assigning the $^{19}$F-NMR signals belonging to the same diFVal residue. All spectra were recorded on a





400 MHz NMR spectrometer with a room temperature probe, except for the short-delay $^1$H,$^{19}$F correlation experiment of GB1-d, which was recorded on a 500 MHz NMR spectrometer equipped with a cryoprobe. The short-delay $^1$H,$^{19}$F correlation experiments were conducted with the delays Δ and δ set to 7 ms and 2.5 ms, respectively. The 1D $^{19}$F-NMR spectra are shown

at the top together with the resonance assignments. The $^1$H chemical shifts of the CH$_2$F groups are between 2.5 and 5 ppm. The cross-peaks with the $^1$H$^β$ resonances are labelled with the $^3J_{HF}$ coupling constants (in Hz, red) derived from the cross-peak intensities. (a) Short-delay $^1$H,$^{19}$F correlation experiment of GB1-1, recorded in 15 h. (b) Same as (a), but for GB1-2, recorded in 13 h. (c) Spectra of GB1-d. The left panel shows the $^{19}$F-detected [$^1$H,$^1$H]-TOCSY spectrum recorded in 19 h with a mixing time of 34 ms. The cross-peaks with H$^β$ resonances of the same residue are connected by purple lines. Stereospecific resonance

assignments are indicated with subscripts, where a 1 or a 2 refers to the $^{19}$F spin attached to the γ$_1$- or γ$_2$-carbon, respectively. The right panel shows the short-delay $^1$H,$^{19}$F correlation experiment recorded in 15 hours. This spectrum was recorded after months of storage, when intact protein mass spectrometry indicated extensive heterogeneity due to partial proteolytic digestion of the flexible C-terminal peptide in our construct (Table S1), including the His$_6$ tag and the TEV cleavage site (Fig. S4).

## 2.4 Stereospecific $^{19}$F-NMR assignments in GB1-d


The stereospecific assignments of the $^{19}$F-NMR resonances of GB1-d were obtained by nuclear Overhauser effects (NOE) observed in a [$^1$H,$^{19}$F]-HOESY spectrum. The strongest NOEs were observed for the buried residue 54 and the weakest for the most highly solvent-exposed residue 21. The NOEs supported the conservation of the dihedral angles χ$_1$ as in the wild-type protein. For example, the *trans* relationship between the H$^α$ and H$^β$ atoms of residue 29 (Fig. 4b) is supported by intense

intraresidual NOEs of both $^{19}$F spins with the H$^α$ atom. In this geometry, the C$^{γ2}$H$_2$F group is closer to the amide proton than the C$^{γ1}$H$_2$F group, which is expected to show a long-range NOE with the C$^{γ1}$H$_3$ group of T18. The HOESY spectrum confirms these NOEs (Fig. 7c). [$^1$H,$^1$H]-NOEs also assist. For example, the side chain conformation of residue 39 (Fig. 4b) predicts an intraresidual [$^1$H,$^1$H]-NOE between the amide proton and the C$^{γ2}$H$_2$F group but not the C$^{γ1}$H$_2$F group. The assignment is confirmed further by an NOE of the γ$_2$-fluorine with the C$^{δ2}$H$_3$ group of L12 in the HOESY spectrum, whereas the γ$_1$-fluorine

makes an NOE with the amide proton of D40. The stereospecific assignments of residue 21 are less straightforward as its $^3J$(H$^α$,H$^β$) coupling is compatible with two different staggered rotamers about the C$^α$–C$^β$ axis. Based on the side-chain conformation shown in Fig. 4b, which is reported both by the X-ray and NMR structures (Gallagher et al., 1994; Juszewski et al., 1999) and supported by the weak intensity of the intraresidual [$^1$H,$^1$H]-NOE between the $^1$H$^β$ and $^1$H$^N$ resonances, the stereospecific assignment of this residue was based on long-range NOEs between the γ$_2$-fluorine and the methyl group of T25,

and between the γ$_1$-fluorine and the α-protons of the glycine residue preceding M1 in our construct (Fig. 6). For residue 54, the HOESY spectrum unambiguously underpins the stereospecific assignments and the conformation of Fig. 4b by intraresidual NOEs between the γ$_1$-fluorine and the amide proton, and between the γ$_2$-fluorine and the α-proton. In addition,



the $\gamma_2$-fluorine shows a well-resolved NOE to the $\varepsilon_1$-proton of the side chain of W43 (Fig. 4c). The stereospecific $^{19}$F-resonance

assignments of GB1-d recapitulate the relative chemical shifts observed in GB1-1 and GB1-2 except for residue 39, which

shows the smallest chemical shift difference between GB1-1 and GB1-2.

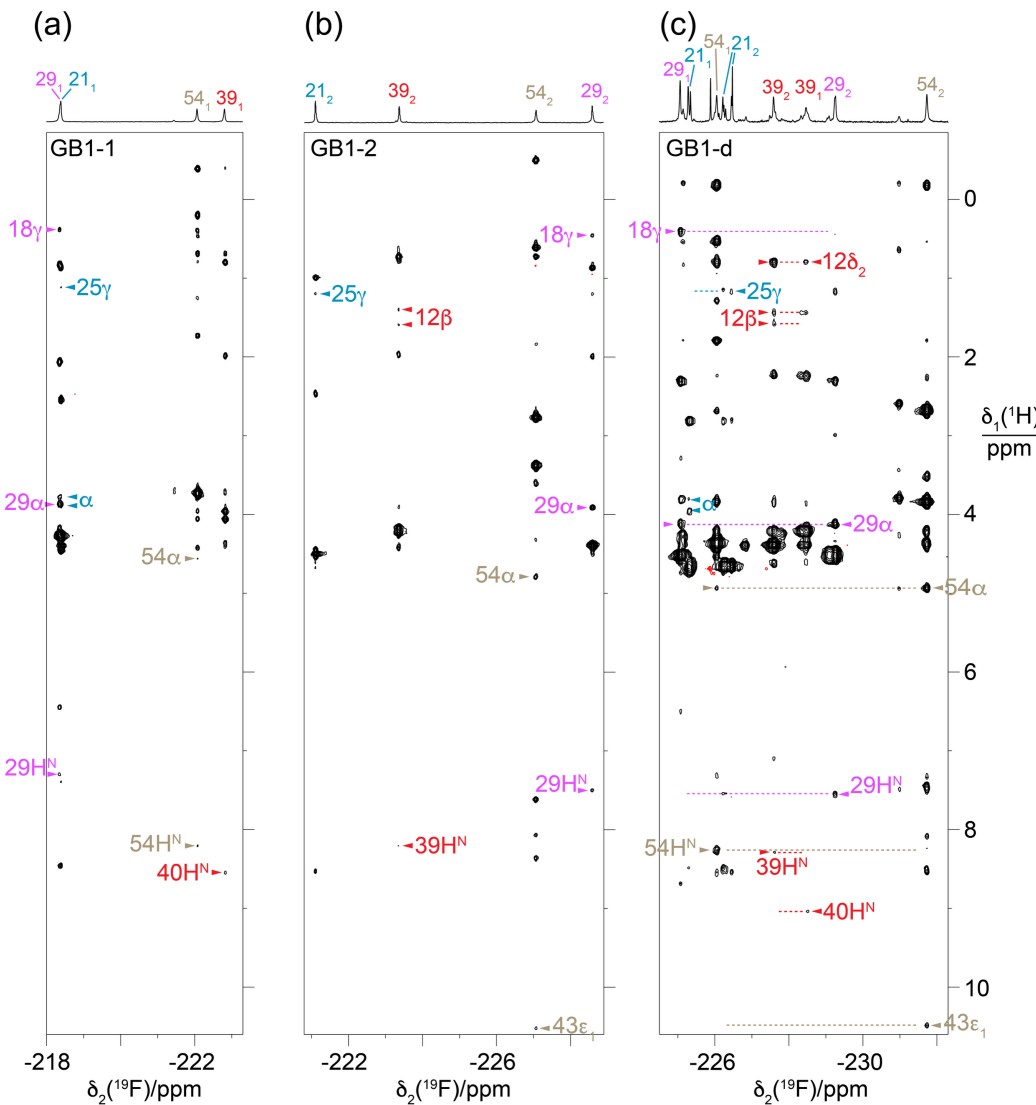

**Figure 6.** HOESY spectra of GB1-1, GB1-2 and GB1-d recorded with a mixing time of 150 ms. The 1D $^{19}$F-NMR spectra are

plotted above the 2D spectra and labelled with the $^{19}$F-resonance assignments, indicating the stereospecific assignments by

subscripts. The different fluorine signals and their cross-peaks are labelled with individual colours. Arrows identify assigned

cross-peaks, with the $^1$H assignments indicated by the residue number and proton type. Two blue arrows labelled $\alpha$ identify

NOEs with the $C^\alpha H_2$ group of the glycine residue preceding M1 in the construct of GB1 used. (a) HOESY spectrum of GB1-





1. Parameters used: $t_{1max}$ = 37.5 ms, $t_{2max}$ = 113 ms, total recording time 36 h. (b) HOESY spectrum of GB1-2 recorded using the same parameters as in (a). (c) HOESY spectrum of GB1-d. The spectrum was recorded on a 500 MHz NMR spectrometer

equipped with a $^{19}$F cryoprobe after the sample had aged as evidenced by new signals and partial proteolytic digestion (Fig. S4). Parameters used: $t_{1max}$ = 18.3 ms, $t_{2max}$ = 133 ms, total recording time 21 h. Dotted horizontal lines link to the locations of putative cross-peaks for the other CH$_2$F group of the diFVal side chains.

## 2.5 Through-space scalar $^{19}$F–$^{19}$F couplings

[$^{19}$F,$^{19}$F]-TOCSY spectra offer a sensitive way of detecting through-space scalar $^{19}$F–$^{19}$F couplings, $^{TS}J_{FF}$ (Orton et al., 2021; Tan et al., 2025). In the 3D structures of GB1 (PDB ID 3GB1 and 1PGA; Juszewski et al., 1999; Gallagher et al., 1994), the C$^{\gamma 1}$ atoms of V39 and V54 are within 4.1 Å. Based on a C–F bond length of 1.40 Å and van der Waals radius of fluorine of 1.47 Å, a direct $^{19}$F–$^{19}$F contact between the FVal1 residues in GB1-1 is conceivable (Fig. 4a). Indeed, the [$^{19}$F,$^{19}$F]-TOCSY spectrum of GB1-1 showed the expected cross-peak, but its intensity is weak (Fig. 7a). As expected for the much greater

distance between the C$^{\gamma 2}$H$_2$F groups (Fig. 4a), GB1-2 did not display a [$^{19}$F,$^{19}$F]-TOCSY cross-peak (Fig. 7b). Interestingly, the [$^{19}$F,$^{19}$F]-TOCSY spectrum of GB1-d showed no cross-peak between the $\gamma_1$-fluorines, instead showing a cross-peak between the $\gamma_1$-fluorine of residue 39 and the $\gamma_2$-fluorine of residue 54 (Fig. 7c). While the solution structure 3GB1 reports the V39 C$^{\gamma 1}$–V54 C$^{\gamma 2}$ distance as 4.4 Å, it is 3.8 Å in the crystal structure 1PGA. This contact thus suggests that the installation of diFVal residues favours the crystal structure conformation, where the dihedral angle $\chi_1$ of V54 is 50º versus 72º in the solution

structure. It is well-known that the fluorine atoms in 1,3-difluoropropane influence the rotamer populations of the CH$_2$F groups. In particular, conformations with intramolecular contacts between the fluorine atoms are unfavourable (Lu et al. 2019; Marstokk and Møllendal 1997; Wu et al. 1998), explaining the absence of intra-residual cross-peaks in the [$^{19}$F,$^{19}$F]-TOCSY spectrum of GB1-d (Fig. 7c).

The $^{19}$F–$^{19}$F contacts reported by the TOCSY spectra must be short-lived, as no corresponding cross-peaks could be

detected in [$^{19}$F,$^{19}$F]-NOESY spectra recorded with longer mixing time and measurement time. In addition, the absence of exchange cross-peaks in the [$^{19}$F,$^{19}$F]-NOESY spectrum of GB1-d (Fig. 7c) support the conclusion that the minor peaks observed in the 1D $^{19}$F-NMR spectrum stem from chemical rather than conformational heterogeneity.

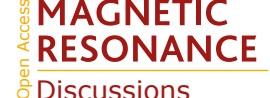

**Figure 7.** $[^{19}F,^{19}F]$-TOCSY spectra showing through-space scalar $^{19}F$–$^{19}F$ couplings, $^{TS}J_{FF}$. Parameters used: 80 ms mixing time (DIPSI-2 mixing), $t_{1max}$ = 4.4 – 5.5 ms, $t_{2max}$ = 113 ms. The 1D $^{19}F$-NMR spectra with the resonance assignments are potted above the spectra. (a) $[^{19}F,^{19}F]$-TOCSY spectrum of GB1-1 recorded in 5.5 h. (b) $[^{19}F,^{19}F]$-TOCSY spectrum of GB1-2 recorded in 2.1 h. (c) $[^{19}F,^{19}F]$-TOCSY spectrum of GB1-d recorded in 5.5 h (top panel) compared with the $[^{19}F,^{19}F]$-NOESY spectrum recorded in about 23 h with a mixing time of 150 ms (bottom panel).







To determine the size of the $^{TS}J_{FF}$ couplings, we measured the relative intensity of the cross-peaks relative to the diagonal peaks in [$^{19}$F,$^{19}$F]-TOCSY spectra recorded with increasing mixing time. This peak ratio indicated that the $^{TS}J_{FF}$ coupling is about 1.5 Hz in GB1-1 and 2.7 Hz in GB1-d (Fig. 8).


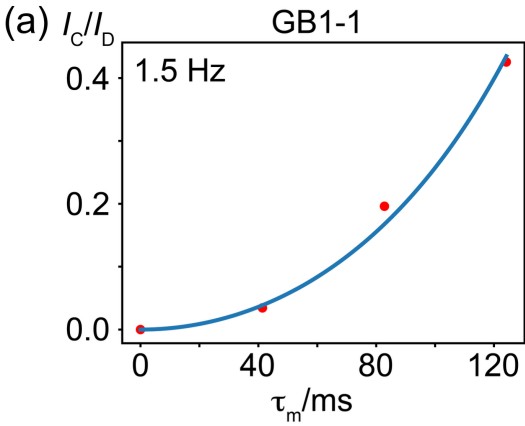

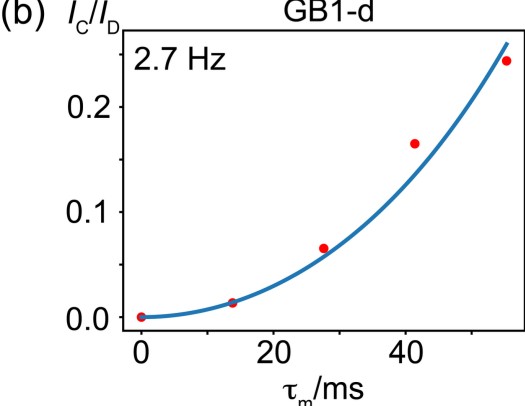

**Figure 8.** Through-space $^{19}$F–$^{19}$F couplings, $^{TS}J_{FF}$, in GB1-1 and GB1-d determined from [$^{19}$F,$^{19}$F]-TOCSY spectra recorded with increasing mixing time $\tau_m$. The ratio of cross-peak ($I_C$) over diagonal-peak ($I_D$) intensity is plotted versus the TOCSY mixing time $\tau_m$. The peak intensities were measured by integrating the respective peak intensities in 1D cross-sections. The
blue curve shows the fit of the function $I_C/I_D = \tan^2(\pi J_{FF}\tau_m)$ (Braunschweiler and Ernst, 1983). The $^{TS}J_{FF}$ coupling determined by a best fit (blue line) is indicated. (a) Coupling between the $\gamma_1$-fluorine atoms of residues 39 and 54 in GB1-1. (b) Coupling between the $\gamma_1$-fluorine of residue 39 and the $\gamma_2$-fluorine of residue 54 in GB1-d. Data recorded at a $^1$H-NMR frequency of 500 MHz.




## 2.6 Rotamers of CH₂F groups: restraints from $^3J_{HF}$ couplings

The 1.3 Å crystal structure of the protein PpiB produced with uniform substitution of valine by FVal1 demonstrated a strong preference of the CH₂F groups for staggered rotamers (PDB ID: 9C5D; Frkic et al., 2024b). The energy barrier between the three rotamers of the CH₂F group in 1-fluoropropane was reported to exceed 4 kcal/mol (Feeney et al., 1996), highlighting the

unfavourable nature of eclipsed conformations. Therefore, it is reasonable to describe the conformations of the CH₂F groups in terms of staggered rotamers with different populations.

The Karplus curve for $^3J_{HF}$ couplings in aliphatic molecules indicates that the $^{19}$F spin couples with the H$^\beta$ atom with coupling constants of about 44 and 8 Hz for torsion angles of 180° (*trans*) and ±60° (gauche), respectively (Williamson et al., 1968; Gopinathan and Narasimhan, 1971). DFT calculations performed for the sixteen FVal1 residues in PpiB indicated

somewhat different coupling constants, namely 30±1 Hz and 9±3 Hz for the $^3J_{HF}$ couplings associated with the *trans* and *gauche* rotamers, respectively (Frkic et al., 2024b). The $^3J_{HF}$ coupling of 38 Hz measured for the γ2-fluorine in position 54 of GB1-d and GB1-2 thus identifies the rotamer that puts the fluorine atom *trans* relative to the $^1$H$^\beta$ atom (Fig. 9).

## 2.7 Rotamers of CH₂F groups: restraints from the γ-effect on $^{13}$C chemical shifts

The $^3J_{HF}$ coupling discriminates between *gauche* and *trans* conformations, but not between the two *gauche* rotamers. To distinguish between the *gauche* rotamers, we use the γ-effect of $^{13}$C chemical shifts in X–C–C–$^{13}$C moieties, where the non-hydrogen substituent X causes a pronounced upfield shift of the $^{13}$C resonance. Empirically, the effect is greatest for fluorine substitutions, which change the $^{13}$C chemical shift by about -6.8 ppm (Günther 2013). The distinction between different rotamers relies on correction terms describing the dependence of the γ-effect on the dihedral angle between the $^{13}$C and $^{19}$F

spin. Conformational corrections of -1.0 ppm and +2.0 ppm have been reported for the torsional angles ±60° and 180°, respectively (Fürst et al., 1990). Using the rotamer nomenclature of Fig. 9, the γ-effect thus identifies the $g^L$ rotamer but does not distinguish between the $g^S$ and $t$ rotamers. The rotamer information gained from the γ-effect on $^{13}$C chemical shifts is therefore complementary to the information obtained from $^3J_{HF}$ couplings.

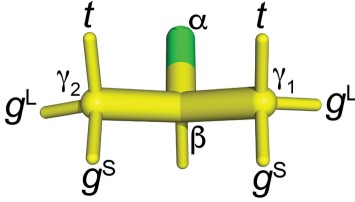

**Figure 9.** Naming convention used in the present work to report the staggered rotamer conformations of CH₂F groups in fluorinated valines. $^3J_{HF}$ is large when the fluorine atom is in the *t* position. In the $g^L$ rotamers, the fluorine atom is in the same plane as the C$^{γ1}$, C$^\beta$ and C$^{γ2}$ atoms. In this conformation of the CH₂F group, a γ-effect produces a relatively large up-field



change of the $^{13}$C chemical shift of the CH$_3$ group (by 6.3 versus 5.1 ppm), and $^3J_{HF}$ between $^1$H$^\beta$ and $^{19}$F is small. For fluorine in the $g^S$ positions, the γ-effect is relatively small and $^3J_{HF}$ is small.


Figure 10 demonstrates how the γ-effect manifests in large up-field changes of the $^{13}$C chemical shifts of the FVal methyl groups in the $^{13}$C-HSQC spectra of GB1-1 and GB1-2 compared with the wild-type protein. The γ-effects range between about 5.8 and 7.4 ppm. In contrast, the $^1$H chemical shift changes of the FVal methyl groups were much smaller, down-field and quite uniform in size apart from the methyl group of residue 54 in GB1-2, which is governed by ring currents from W43. Even the methyl groups of residue 54 changed their $^1$H chemical shifts by less than 0.25 ppm.

In principle, the γ-effects displayed by the $^{13}$C chemical shifts are modulated by the chemical environment such as the presence or absence of fluorine atoms in nearby residues. Importantly, however, the $^{13}$C chemical shifts in GB1-1 and GB1-2 appear to respond only little to the presence of nearby FVal residues as shown by the methyl groups of alanine, leucine and threonine located within NOE distance of the fluorovaline side chains, which changed their $^{13}$C chemical shifts by less than 0.3 ppm (Fig. 10). This indicates that the conformational correction term of the γ-effect easily exceeds the impact on $^{13}$C chemical shifts arising from changes in the local environments by FVal-labelling.







**Figure 10.** Methyl region of the $^{13}$C-HSQC spectra of GB1-1 and GB1-2 superimposed onto the spectrum of wild-type GB1. The spectra were recorded at natural isotopic abundance using a $^1$H NMR frequency of 800 MHz. (a) Spectrum GB1-1 (red) superimposed on the spectrum of wild-type GB1 (black). The cross-peaks of the C$^{\gamma 2}$H$_3$ groups of valine residues in wild-type GB1 are labelled with the residue number. Arrows point to the corresponding cross-peaks in GB1-1. The stereospecific assignments in the wild-type protein were reported by Goehlert et al. (2004). (b) Same as (a), but for GB1-2 (blue spectrum), showing the change in chemical shifts for the valine C$^{\gamma 1}$H$_3$ groups.





**2.8 Rotamers of CH₂F groups: restraints from $^3J_{FC}$ coupling constants**

If the magnitude of the γ-effect is significantly determined by the rotamer states of the CH₂F groups, it should correlate with
the $^3J_{FC}$ couplings of the CH₃ groups of the FVal residues. Therefore, we measured the $^3J_{FC}$ couplings of the CH₃ groups of the
FVal residues in GB1-1 and GB1-2. As the samples were at natural isotopic abundance, we used the constant-time $^{13}$C-HSQC
difference experiment (Grzesiek et al., 1993) for best sensitivity. The coupling constants measured varied between 5.0 and 8.6
Hz. The largest coupling constants were measured for the FVal1 residues 21 and 29 in GB1-1 (8.6 and 8.3 Hz, respectively;
Fig. 11). As expected, the up-field changes of the $^{13}$C chemical shifts of the FVal methyl groups, $\Delta\delta(^{13}C)$, correlate with the
$^3J_{FC}$ couplings (Fig. 12a).

**2.9 Density-functional theory (DFT) calculations**

For an independent underpinning of the γ-effect, we performed DFT computations on FVal1 residues in PpiB, for which a
high-resolution crystal structure is available with FVal1 residues at sixteen sites (Frkic et al., 2024b). The calculations indicated
that the presence of the $\gamma_1$-fluorine atom changes the chemical shift of the $^{13}C^{\gamma 2}$ spin by -6.3±0.4 ppm, if the intervening dihedral
angle for rotation about the $C^\beta$–$C^{\gamma 1}$ bond is 169±6 degrees (i.e., the range of angles found following energy minimization of
the FVal1 side-chain conformations). In contrast, the $^{13}C^{\gamma 2}$ chemical shift was calculated to change only by -5.1±0.2 ppm and
-5.1±0.3 ppm for the dihedral angles -61±6 degrees and 60±2 degrees, respectively.

using a 1-fluoro-2-methylpropane model compound with a systematic change of the torsion angle τ (Fig. 12b). Two
sets of torsion-dependent calculations were performed to evaluate the sensitivity of the DFT calculations towards minor
changes in bond lengths and bond angles: one where *only* the relevant torsion angle was varied and all other atomic positions
were fixed (Fig. 12b and S7), and the second where all other degrees of freedom were relaxed for each specified torsion angle
(Fig. S8). Both sets of calculations confirmed that the γ-effect is significantly larger for the torsion angle τ = 180º than for τ
= ±60º (Fig. 12b, S7 and S8). The only notable difference in the two sets of calculations is that the $^3J_{FC}$ coupling constant for
τ = 0º was smaller with optimisation of the other molecular degrees of freedom (Fig. S7 and S8).

**MAGNETIC RESONANCE**
Open Access Discussions

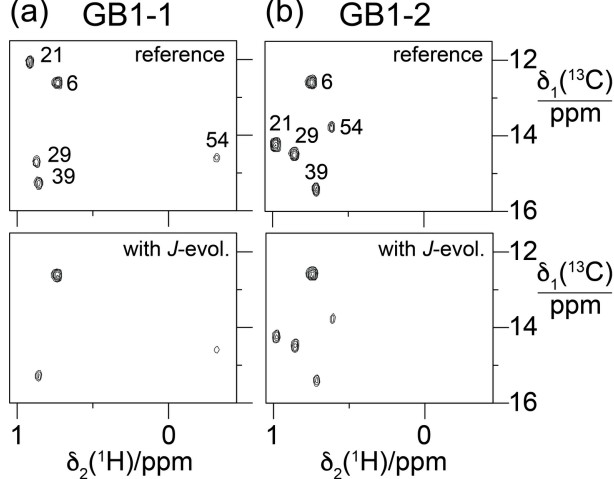

**Figure 11.** Constant-time $^{13}$C-HSQC difference experiments for measuring $^3J_{FC}$ couplings between $^{19}$F and the $^{13}$C spin of the CH$_3$ group (Grzesiek et al., 1993). Spectra with and without $J_{FC}$ coupling evolution were recorded in an interleaved manner. The experiment used the Bruker pulse program hsqcctetgpjclr with adaptation for $J_{FC}$ instead of $J_{CC}$ couplings. Parameters used: $t_{1max}$ = 19 ms, $t_{2max}$ = 82 ms, total recording time 10 h. Spectra recorded on a 500 MHz NMR spectrometer equipped with a $^1$H/$^{19}$F/$^{13}$C cryoprobe. (a) Spectra obtained with GB1-1. The top panel shows the reference spectrum recorded with refocused $J_{FC}$ coupling, whereas $J_{FC}$ couplings evolved for a period of 57.5 ms in the spectrum shown in the bottom panel. The cross-peak intensities are conserved for Ile6, which does not couple to $^{19}$F. (b) Same as (a) but for GB1-2.

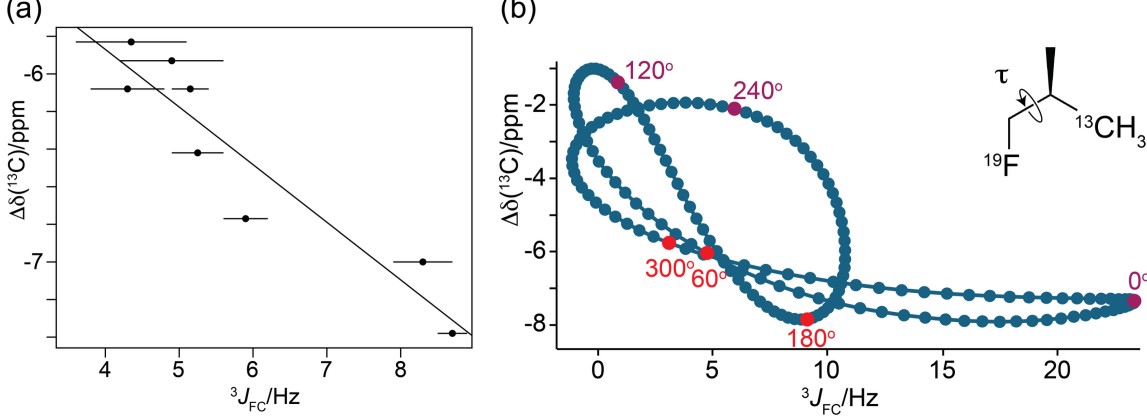

**Figure 12.** γ-effects of the $^{13}$C chemical shifts of FVal methyl groups generated by the fluorine atom in the CH$_2$F group, $\Delta\delta(^{13}$C). (a) Correlation of $\Delta\delta(^{13}$C) with the $^3J_{FC}$ coupling constants. $\Delta\delta(^{13}$C) is calculated as the $^{13}$C chemical shift of FVal-labelled GB1 minus the $^{13}$C chemical shift of the corresponding valine residue in wild-type GB1. Error bars indicate uncertainties in cross-peak intensities. The straight line was determined by a least-squares fit. (b) Correlation of $\Delta\delta(^{13}$C) with the $^3J_{FC}$ coupling constants obtained by DFT calculations for (2R)-1-fluoro-2-methylpropane(3-$^{13}$C) where *only* the torsion



angle is varied and all other coordinates are fixed. The vertical axis reports $\Delta\delta(^{13}C)$ relative to the methyl $^{13}C$ chemical shift calculated for isobutane. The insert shows the chemical structure of the 1-fluoro-2-methylpropane compound. For the torsion angle $\tau = 0$, the $^{19}F$ and $^{13}C$ nuclei are *syn* with respect to the intervening carbons and in the same plane. The plot highlights the points calculated for the torsion angles of the staggered conformations (red) and the energetically unfavourable eclipsed conformations (magenta).

### 2.10 Rotamers determined by $^3J_{HF}$ couplings, $^3J_{FC}$ couplings and the γ-effect on $^{13}C$ chemical shifts

According to the Karplus curve, the FVal residues possess larger $^3J_{FC}$ constants for the $g^L$ than the $g^S$ and $t$ rotamers. In rigid organic molecules containing C-C-C-F moieties, $^3J_{FC}$ values of about 10 Hz were reported for dihedral angles of 180° (Barfield et al., 1978). The smaller $^3J_{FC}$ constants measured for GB1-1 and GB1-2 thus suggest that the CH₂F groups of the FVal residues are subject to some degree of motional averaging. Nonetheless, our data show evidence of preferential rotamer populations.

For example, the $C^{\gamma 2}H_2F$ group of residue 54 in GB1-2 and GB1-d clearly populates the $t$ rotamer as evidenced by the large $^3J_{HF}$ coupling and up-field $^{19}F$ chemical shifts. As expected for this conformation, the γ-effect of the $^{13}C^{\gamma 1}$ spin in GB1-2 is relatively small (-5.93 ppm versus the average of -6.4 ppm; Table 1; Fig. 10b). In addition, the $^3J_{FC}$ coupling of the $C^{\gamma 2}H_2F$ group of residue 54 in GB1-2 is relatively small as expected for a dihedral angle of about -60° (5 Hz; Table 1).

The data are less decisive regarding the rotamer preference of the $C^{\gamma 1}H_2F$ group of residue 54. In GB1-1, its $^3J_{HF}$ coupling is relatively small, suggesting a $g$ rotamer, but it is not as small as in GB1-d (Table 1). Its $^{19}F$ chemical shift is low-field, but not as much as for other residues, suggesting an admixture of the $t$ rotamer. The γ-effect is relatively small, but not as small as expected if only $g^S$ or $t$ rotamers were populated. However, the very small $^3J_{FC}$ coupling in GB1-1 (4.3 Hz) excludes a significant population of the $g^L$ rotamer. In summary, the $g^S$ rotamer appears preferred. A better fit would have to consider additional rotamers and deviations from the perfectly staggered conformations.

The rotamer analysis of other CH₂F groups similarly does not indicate single rotamers, but preferential populations can be identified. For example, the largest γ-effects of FVal residues were observed for the $^{13}C^{\gamma 2}$ spins of residues 21 (-7.38 ppm) and 29 (-7.00 ppm) in GB1-1. These shifts, as well as relatively large $^3J_{FC}$ couplings, suggest that the respective FVal1 residues preferentially populate the $g^L$ rotamer. As expected for this rotamer, the $^3J_{HF}$ couplings of both FVal1 residues are small, their $^{19}F$ chemical shifts are down-field (Fig. 2a) and their $^3J_{FC}$ couplings are relatively large (Table 1).

Table 1 and Fig. 13 summarize the preferential rotamers for the FVal residues in GB1-1 and GB1-2, derived from $^3J_{HF}$ couplings, $^3J_{FC}$ couplings and the γ-effect of the $^{13}C$ chemical shift of $C^{\gamma}H_3$ groups with the assumption of staggered rotamers.





**Table 1.** Determination of $CH_2F$ rotamers from $^3J_{HF}$ couplings, $^{19}F$ chemical shifts $^{13}C$ chemical shift changes and $^3J_{FC}$ coupling constants[a]

| Protein and residue | $^{19}F$ chemical shift (ppm) | $^3J_{HF}$ (Hz) | Preferred rotamers based on $^3J_{HF}$[b] | $^{13}C$ γ-effect (ppm)[c] | $^3J_{FC}$ (Hz)[d] | Preferred rotamers[e] |
|---|---|---|---|---|---|---|
| **GB1-1** | | | | | | |
| 21 | -218.39 | 19 | $g$ | -7.38 | 8.7 | $g^L$ |
| 29 | -218.36 | 13 | $g$ | -7.00 | 8.3 | $g^L$ |
| 39[b] | -222.80 | 18 | $g$ | -6.42 | 5.2 | $g^S$ |
| 54 | -222.06 | 17 | $g$ | -6.08 | 4.3 | $g^S$ |
| **GB1-2** | | | | | | |
| 21 | -221.12 | 22 | $g, t$ | -6.77 | 5.9 | $g^L, t$ |
| 29 | -228.59 | 26 | $t, g$ | -5.83 | 4.4 | $t, g^S$ |
| 39 | -223.38 | 21 | $g, t$ | -6.08 | 5.2 | $g^S, t$ |
| 54 | -227.07 | 38 | $t$ | -5.93 | 4.9 | $t$ |
| **GB1-d** | | | | | | |
| $21_1$ | -226.27 | 27 | $t, g$ | | | |
| $21_2$ | -225.34 | 17 | $g$ | | | |
| $29_1$ | -225.16 | 21 | $g, t$ | | | |
| $29_2$ | -229.30 | 25 | $t, g$ | | | |
| $39_1$ | -228.67 | 19 | $g$ | | | |
| $39_2$ | -227.60 | 26 | $t, g$ | | | |
| $54_1$ | -226.13 | 12 | $g$ | | | |
| $54_2$ | -231.73 | 38 | $t$ | | | |

[a] The $^3J_{HF}$ couplings were measured by line fitting in 1D $^{19}F$-NMR spectra and short-delay $^1H,^{19}F$ correlation experiments (Tan et al., 2024). For the $C^{γ2}H_2F$ group of residue 54, the $^3J_{HF}$ coupling was measured from the peak splittings observed for the $C^{γ2}H_2$–$H^β$ NOESY cross-peaks.

[b] Where two rotamers are listed, the first rotamer appears more highly populated than the second rotamer which is more highly populated than the third rotamer. In the case of residue 39 in GB1-1, the data suggest a preference for the $g^S$ rotamer while a preference of the $g^L$ versus $t$ rotamer is difficult to discern.

[c] The data for GB1-1 pertain to the $C^{γ2}H_3$ group and the data for GB1-2 to the $C^{γ1}H_3$ group. The γ-effect on $^{13}C$ chemical shifts, $Δδ(^{13}C)$, is reported as $δ(^{13}C)$ of the FVal-labelled protein minus the $δ(^{13}C)$ value of the wild-type protein.

[d] Coupling constant of $^{19}F$ to the carbon of the $CH_3$ group measured in a spin-echo difference experiment.



ᵉ Based on $^3J_{HF}$, $\Delta\delta(^{13}C)$ and $^3J_{FC}$. See Fig. 9 for the nomenclature used.

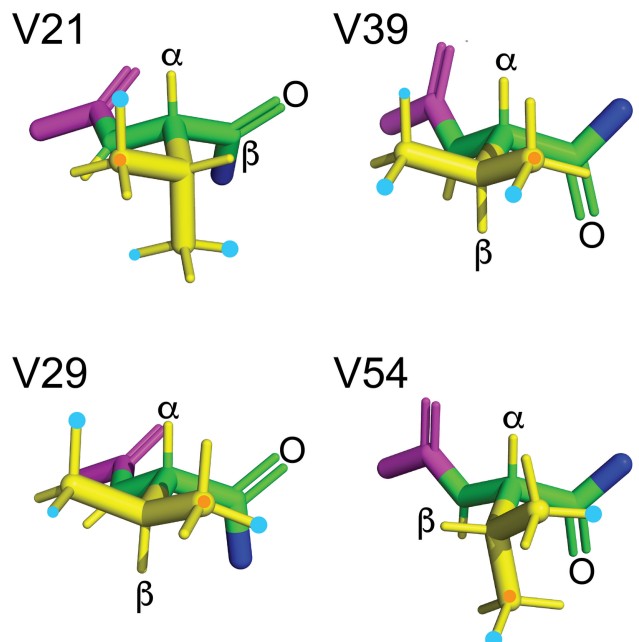

**Figure 13.** Rotamers preferentially populated by fluorovaline residues in GB1-1 and GB1-2. The four valine side chains are shown in the conformation of the solution structure 3GB1 with the methyl groups rotated into fully staggered conformations. Cyan dots mark the preferred locations of fluorine atoms in GB1-1 and GB1-2 based on the data of Table 1. When the data

clearly point to the population of more than a single rotamer, the most highly populated rotamer is identified by a larger dot. An orange dot marks the $\gamma_1$ carbon. The backbone atoms N, $C^\alpha$, C and O are shown in green, the $H^N$, $H^\alpha$ and side chain atoms in yellow, the $C^\alpha$, C and O of the previous residue in magenta and the nitrogen of the following residue in blue. The double bond of each carbonyl group is shown by two lines and the carbonyl oxygen of each valine residue is labelled.

Assigning preferential rotamers in GB1-d is more difficult, as the diFVal residues contain no $CH_3$ group, which makes $^3J_{FC}$ coupling measurements difficult. Furthermore, the $^{13}C$ chemical shifts of the $CH_2F$ groups are under the influence of the directly bonded fluorine atoms. While the $\gamma$-effect on $^{13}C$ chemical shifts should also be sensed by the $\alpha$-carbons, a qualitative analysis of GB1-1 showed that the $^{13}C^\alpha$ chemical shifts of the FVal residues varied much more than expected for the $\gamma$-effect. Reliable restraints were therefore limited to the $^3J_{HF}$ couplings. For residue 54, the $^3J_{HF}$ couplings suggest that the diFVal side

chain preferentially populates the same rotamers as the FVal1 and FVal2 analogues, except that the smaller $^3J_{HF}$ coupling of the $\gamma_1$-fluorine suggests a greater preference for the $g^S$ rotamer.

      In contrast, diFVal in position 21 showed a larger $^3J_{HF}$ coupling for the $\gamma_1$-fluorine than GB1-1 (Table 1), suggesting a change in preferred rotamer to the $t$ rotamer. Notably, a diFVal residue maintaining the $g^L$ rotamers indicated by the FVal1





and FVal2 data would lead to an all-*trans* conformation of the F–$C^{\gamma 1}$–$C^{\beta}$–$C^{\gamma 2}$–F moiety, which has been shown to be
energetically unfavourable in 1,3-difluoropropane (Marstokk and Møllendal 1997; Wu et al. 1998; Lu et al. 2019). Likewise,
maintaining the $g^S$ rotamers identified for residue 39 as most highly populated in GB1-1 and GB1-2 would bring the two
fluorines into close proximity in GB1-d, which is energetically unfavourable. In this case, the $\gamma_2$-fluorine changes its rotamer
preference GB1-d in favour of the *t* rotamer as indicated by an increased $^3J_{HF}$ coupling constant (Table 1). Despite the larger
$^3J_{HF}$ coupling, however, the $^{19}$F-NMR signal of the $\gamma_2$-fluorine is down-field of the $\gamma_1$-fluorine signal in GB1-d, in apparent
violation of the $\gamma$-gauche effect. Notably, however, the chemical environment of the $\gamma_1$-fluorine changes in GB1-d as evidenced
by the switch in $^{TS}J_{FF}$ coupling from the $\gamma_1$-fluorine of residue 54 to the $\gamma_2$-fluorine (Fig. 7a and c), compromising the
interpretation of the $^{19}$F chemical shifts.

The rotamer assignments of the $C^{\gamma 1}H_2F$ groups of residues 39 and 54 agree with the $^{TS}J_{FF}$ coupling observed in the
[$^{19}$F,$^{19}$F]-TOCSY spectrum of GB1-1 (Fig. 7a), as the NMR structure 3GB1 of the wild-type protein suggests that the shortest
possible $^{19}$F–$^{19}$F distance is between the $\gamma_1$-fluorines of these residues if both populate the $g^S$ rotamer. These rotamers are
indeed highly populated (Table 1). In contrast to the structure 3GB1, the crystal structures 1PGA, 1PGB (Gallagher et al.,
1994) and 2QMT (Frericks Schmidt et al., 2007) show the $\gamma_1$-carbon of Val39 closer to the $\gamma_2$-carbon of Val54 (3.8 Å versus
4.2 Å). The additional fluorine atoms in GB1-d thus seem to effect the small structural change towards the conformation
observed in the single crystal (Fig. 7c).

## 2.11 Fluorines near carbonyl carbons

The rotamer analysis shows that the FVal2 residues in positions 21 and 54 and the FVal1 residue in position 29 preferentially
populate a rotamer that positions the fluorine atom in close proximity of the carbonyl carbon of the same residue (Fig. 13).
This conformation, which results in a C–F distance as short as 2.85 Å, has been noted previously in the crystal structure of
PpiB made with FVal1 (Frkic et al., 2024b). It suggests a favourable interaction between the negatively polarized fluorine and
the positively polarized carbonyl carbon.

To explore whether these close fluorine–carbonyl carbon interactions are manifested in the $^{13}$C chemical shifts of the
carbonyl carbons, we recorded HNCO experiments of GB1-1, GB1-2 and wild-type GB1 at natural $^{13}$C abundance. The wild-
type protein was uniformly labelled with $^{15}$N by conventional expression in vivo, while the samples containing FVal1 or FVal2
were produced by cell-free protein synthesis from amino acids, where the amino acids following the valine sites in the amino
acid sequence (asparagine, aspartic acid, phenylalanine and threonine) were labelled with $^{15}$N. This selective labelling approach
allowed the resonance assignment from the first 2D plane of the HNCO experiment correlating $^{13}$C chemical shifts with
backbone amide protons (Fig. S5).

Most notably, the GB1-1 and GB1-2 samples showed significant up-field changes in the $^{13}$C chemical shifts of the
carbonyl groups of the FVal1 and FVal2 residues, with GB1-2 showing the largest $^{13}$C chemical shift changes. The effect does



not correlate with the distance between $^{19}F$ and $^{13}C$ spins, however, as large internuclear distances still produced large chemical shift changes as shown, for example, by residues 29 and 39 in GB1-2. In contrast, residue 54 showed by far the smallest change in $^{13}C$ chemical shift. Therefore, the changes in $^{13}C$ chemical shifts are not a simple function of spatial proximity.

To explore whether fluorine changes chemical shifts mostly via through-bond or through-space effects, we recorded $^{15}N$-HSQC spectra (Fig. S6). The amide cross-peaks of all fluorinated valine analogues shifted significantly, with all amide protons shifting down-field, while the amide nitrogens shifted up-field except for residue 21 in GB1-2. The shifts observed for the diFVal residues in GB1-d corresponded by and large to the sum of the chemical shift changes observed for GB1-1 and GB1-2. This was also observed for the most strongly shifting cross-peaks of non-valine residues. This cumulative effect of the fluorination appears to be insensitive to the rotamer populations of the CH$_2$F groups. Some of the largest $^1H$ chemical shift

changes were observed for the amide protons of residues following the valine sites. Comparing the chemical shift changes of residues 29 and 39, which feature the same torsion angle $\chi_1$ between the α- and β-carbons (Fig. 13), it is interesting to note the similarity in chemical shift changes of their own amide cross-peaks as well as the amide cross-peaks of the following residues. As through-bond effects, these effects would be active six bonds from the $^{19}F$ spin, but the conservation of the $\chi_1$ angle could also entail a conserved structural perturbation.


## 3 Discussion

### 3.1 Effects of fluorinated valines on protein fold stability

Highly conserved $^1H$ chemical shifts confirm that the 3D structure of GB1 is fully maintained when all four valine residues are replaced by fluorinated valine analogues, but their presence destabilizes GB1 with respect to heat denaturation. Owing to

the shorter side chain, the CH$_2$F groups of the fluorinated valine analogues are closer to the backbone than those of correspondingly fluorinated leucine analogues, leaving the CH$_2$F groups with fewer options to be accommodated in the 3D protein fold. Nonetheless, the melting temperatures of the GB1 variants made with fluorinated valines did not decrease much more than for GB1 samples made with fluorinated leucine (Tan et al., 2025). We previously noted that the substitution of valine for FVal1 in the protein PpiB, which contains sixteen valine residues, reduced the melting temperature by only 11 °C

(Frkic et al., 2024b), i.e., only little more than the substitution of five leucine residues by fluorinated analogues (Tan et al., 2024).

        The rotamers preferred by the CH$_2$F groups of FVal1 and FVal2 residues often position the fluorine atom next to the carbon of backbone carbonyl groups (Fig. 13) despite the nominal van der Waals radii suggesting steric hindrance. Examples for this conformation have been observed previously in the high-resolution crystal structure of FVal1-labelled PpiB (Frkic et

al., 2024b). It has been argued that the standard van der Waals radii are not appropriate for sp$^2$ carbons bound to electron-withdrawing moieties (Kruse et al., 2020). Oxygen atoms have been reported to form weakly stabilizing n→π* bonds with

**MAGNETIC RESONANCE**
Discussions

carbonyl carbons (Bartlett et al., 2010) or simply engage in a favourable dipole–dipole interaction (Worley et al., 2012). The experimental results of the present work indicate that the interaction between fluorine and a carbonyl carbon may promote rotamers with short fluorine–carbon distances but it is insufficient to lock the $CH_2F$ group into a single rotamer. The interaction
does not manifest in unusually large changes in $^{13}C$ chemical shifts of the carbonyl groups.

### 3.2 Preferential rotamer populations of $CH_2F$ groups

Each $CH_2F$ group can access three different staggered rotamers, which are separated by energy barriers of about 4–5 kcal/mol (Feeney et al., 1996). As shown by the 1.3 Å crystal structure of FVal1-labelled PpiB, deviations from perfectly staggered
conformations are small, but the protein environment rarely confines the fluorine atoms to single rotamers (Frkic et al., 2024b). The present work confirms the population of multiple rotamers in GB1 both for solvent-exposed and buried $CH_2F$ groups. A strong preference for single rotamers was detected only for residue 54, which is the most deeply buried valine residue in wild-type GB1. Nonetheless, rotamer preferences can also be discerned for highly solvent-exposed $CH_2F$ groups such as in residue 21. Interestingly, the HCl salts of the free amino acids dissolved in methanol already show evidence of preferential rotamer
populations, as the $^{3}J_{HF}$ and $^{3}J_{FC}$ couplings and $^{19}F$ chemical shifts differ between the FVal1 and FVal2 amino acids. The values reported for FVal1 and FVal2, respectively, are 20.5 Hz and 15.5 Hz for $^{3}J_{HF}$, -224.55 ppm and -222.17 ppm for $\delta(^{19}F)$, and 7.5 Hz and 7.3 Hz for the $^{3}J_{FC}$ coupling of the $CH_3$ group (Maleckis et al., 2022). Most notably, the $^{3}J_{HF}$ couplings and $^{19}F$ chemical shifts correlate as expected for the γ-gauche effect (Feeney et al., 1996; Tan et al., 2024). The correlation also holds for diFVal, where the $^{3}J_{HF}$ couplings are 20.8 Hz and 23.6 Hz for the $^{19}F$-NMR signals at -223.93 ppm and -225.72 ppm,
respectively (Maleckis et al., 2022).

### 3.3 Through-space $J_{FF}$ couplings

As in previous work with fluorinated amino acids (Kimber et al., 1978; Orton et al., 2021; Tan et al., 2025), $^{TS}J_{FF}$ couplings between fluorinated valine analogues were readily observed by $[^{19}F,^{19}F]$-TOCSY experiments. The couplings are much smaller
than the $^{TS}J_{FF}$ coupling of 21 Hz reported for 6-fluorotryptophan-labelled dihydrofolate reductase, where the fluorine atoms are held in a rigid structure relative to each other (Kimber et al., 1978). Therefore, the $^{TS}J_{FF}$ couplings detected in the present work are evidence of direct contacts but not of attractive forces between the fluorine atoms, despite the generally hydrophobic nature of carbon-bonded fluorine. To the contrary, the polarity of the C–F bonds may well discourage direct $^{19}F$–$^{19}F$ contacts, just as bond polarities restrict the accessible rotamer combinations in diFVal residues, hindering the detection of intra-residual
$[^{19}F,^{19}F]$-TOCSY cross-peaks between the $CH_2F$ groups of diFVal.





### 3.4 γ-effect on $^{13}$C chemical shifts

The γ-effect and its dependence on the torsion angle of the intervening atoms have been known for a long time (Fürst et al., 1990). A γ-effect has also been reported for the $^{13}$C-chemical shift of δ-methyl groups of leucine residues, when the $C^\alpha$–$C^\beta$

and $C^\gamma$–$C^\delta$ bonds form a dihedral angle of 180° about the $C^\beta$–$C^\gamma$ bond (MacKenzie et al., 1996). $^{19}$F spins produce the largest γ-effects (Günther 2013). To the best of our knowledge, the present work is the first experimental demonstration of the dihedral angle dependence of the $^{19}$F-driven γ-effect. The γ-effect also prevails for singly fluorinated leucine analogues in GB1, where the smallest Δδ($^{13}$C) values have been detected for the methyl groups of the buried residue 5, which is characterised by pronounced rotamer preferences of the CH$_2$F groups (Tan et al., 2025). As the difference between the $^{13}$C chemical shifts of

canonical valine residues and their fluorinated analogues, Δδ($^{13}$C), is influenced also by any structural adjustments caused by the fluorine atoms, rendering the γ-effect a less straightforward parameter for interrogating dihedral angles than $^3J_{FC}$ couplings.

### 3.5 Using the γ-effect in site-specific probes

As demonstrated by the current results, the hydrogen-fluorine substitution in a methyl group does not fully stop the rotation of

the resulting CH$_2$F group, even if it is buried in the core of a structurally stable protein. This highlights the ease with which fluorine atoms can be accommodated in the protein structure. As the γ-effect translates changes in the population of different rotamers into a significant change of the $^{13}$C chemical shift of the remaining CH$_3$ group, a fluorinated isopropyl probe could be used as a sensor of ligand binding, which can be interrogated with a spectrometer that is not equipped for $^{19}$F NMR.

### 3.6 Sequence-specific assignment of $^{19}$F NMR resonances


As chemical shift anisotropy adds a prominent relaxation mechanism for $^{19}$F spins, the broad $^{19}$F-NMR signals encountered for large proteins often call for resonance assignment by site-directed mutagenesis. Fortunately, much information can be drawn from the short-delay $^1$H,$^{19}$F correlation experiment, which provided excellent sensitivity for linking $^{19}$F and $^1$H chemical shifts even in the 19 kDa protein PpiB containing fluorinated leucine residues (Tan et al., 2024). The present study shows that

different fluorinated valine residues in GB1 maintain the same relative order of average $^1$H chemical shifts in the CH$_2$F groups as the valine CH$_3$ groups in the wild-type protein. The same applies for the H$^\beta$ atoms that are also detected by the short-delay $^1$H,$^{19}$F correlation experiment. Inspection of the NMR spectra reported previously of GB1 (Tan et al., 2025) and PpiB (Tan et al., 2024) containing fluorinated leucine residues likewise reveals full conservation of the relative ordering of the $^1$H chemical shifts of the H$^\gamma$ atoms as in the wild-type protein. In GB1, even the $^1$H chemical shifts of the $C^\delta$H$_2$F groups follow the average

chemical shift ordering of the methyl groups in the wild-type protein. This greatly aids the $^{19}$F-resonance assignments if the

[1]H assignments of the wild-type protein are available. Alternatively, genetic encoding of amino acids differing from their parents by a single H-to-F substitution has recently become possible in vivo (Qianzhu et al., 2022; 2024; 2025) and may become possible also for FVal and FLeu residues.

### 3.7 Comparison of the NMR properties of fluorinated valines and fluorinated leucines

In previous work, we substituted leucines for singly and doubly fluorinated analogues in GB1 (Tan et al., 2025) and PpiB (Tan et al., 2024). The leucine analogues are very similar to FVal1, FVal2 and diFVal, except for the presence of an additional $CH_2$ group between the α-carbon and the isopropyl group, which adds flexibility to the amino acid side chains. As expected for increased side chain mobility, the transverse [19]F relaxation rates of diFLeu residues measured in GB1 were slower than for the diFVal-labelled sample. Furthermore, some of the [19]F NMR signals of PpiB made with FVal1 were very broad, suggesting exchange broadening. In contrast, none of the FLeu1 or FLeu2 signals showed similarly severe line broadening effects and more uniform peak heights were obtained with the singly fluorinated leucine variants than with diFLeu, confirming the importance of local side chain mobility for narrow [19]F-NMR signals. In all examples, the signal heights were lowest for the most deeply buried $CH_2F$ groups.

### 4 Conclusions

Proteins prepared with either FVal1, FVal2, FLeu1 or FLeu2 produce [19]F-NMR spectra with extraordinary spectral widths and are only little destabilized compared with their wild-type parents. The recent commercial availability of these amino acids greatly advances the accessibility of the corresponding proteins. The conservation of relative [1]H chemical shifts associated with the conservation of the side chain conformations opens a convenient route to assigning the [19]F NMR spectra. The pronounced sensitivity of $CH_2F$ groups towards changing protein environments renders them excellent NMR probes of, e.g., ligand binding.

**Data availability.** The NMR data are available at https://doi.org/10.5281/zenodo.17082808 (Otting, 2025).

**Author contributions.** EHA prepared the protein samples and performed 1D NMR measurements. NFC performed the DFT calculations. AM synthesised fluorinated valine analogues. GO coordinated the project, performed the 2D NMR measurements and prepared the final manuscript and figures.





**Acknowledgements.** We thank Drs Roger Amos, Rika Kobayashi and Michael Collins for preliminary DFT calculations of [19]F chemical shifts and Prof. Martin Scanlon for access to the 500 MHz NMR spectrometer with [1]H/[19]F/[13]C-cryoprobe.

**Financial Support.** This research has been supported by the Australian Research Council (grant no. DP230100079) and the Australian Research Council Centre of Excellence for Innovations in Peptide and Protein Science (grant no. CE200100012).

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
