# Peer review of "γ-effects identify preferentially populated rotamers of CH2F groups: side-chain conformations of fluorinated valine analogues in a protein"

_Magnetic Resonance, 2025_

## Author Comment (AC3)

Thank you for the careful read!

1. **Additional species in GB1-d (Fig. 2c):**

   Could you expand the discussion regarding the presence of additional species observed in Fig. 2c? For instance, why are these species not present in GB1-1 and GB1-2? You write, "Therefore, the low-intensity peaks in the 19F-NMR spectrum of GB1-d most likely originate from the species with one valine and three diFVal residues." Does this imply that the absence of one diFVal residue causes a significant change in the chemical shifts of the other 19F-labeled valines? Additionally, could you specify which residue is missing, since the small number of low-intensity peaks suggests that not all possible combinations are present in solution?

**Response:** We observed similar low-intensity $^{19}$F-NMR signals in our previous work with 5,5'-difluoroleucine (diFLeu; Tan et al., 2025). These signals became the dominant signals in a sample prepared with a mixture of diFLeu and Leu to yield protein containing single diFLeu residues. The resonance assignments of this sample (referred to as GB1-dd) showed that the dilution of the diFLeu residues by Leu most strongly affected the $^{19}$F chemical shifts of diFLeu7 (by about 3 ppm). Leu7 is located between Leu5 and Leu12. The $^{19}$F chemical shifts of diFLeu in positions 5 and 12 changed by less than 1 ppm. Furthermore, GB1 made with diFLeu gave no hint of chemical exchange between main and weak peaks.

Based on this we believe that the $^{19}$F chemical shifts of diFVal are similarly sensitive to the presence or absence of fluorine atoms at other valine sites. Val39 and Val54 are next to each other, so the main peaks of these two residues should be accompanied by four weak signals of equal intensity at chemical shifts quite different from the protein containing 100% diFVal. For example, the lonely weak signal at -231 ppm almost certainly stems from diFVal in position 54 with valine in position 39 (Fig. 5c). Counting all the weak signals and peak shoulders in the 1D $^{19}$F-NMR spectrum, we can account for a total of eight weak signals, which suggests that the identity of the nearest diFVal-versus-Val neighbour can matter also over greater distances. Unfortunately, as we don't have any diFVal left, we cannot assign all the weak peaks.

The abundance of protein containing two diFVal and two Val residues is insufficient for detection by NMR (Fig. S4a).

Weak signals are less prominent in GB1-1 and GB1-2, because the singly fluorinated valine analogues are less easily outcompeted for incorporation into the

protein by traces of canonical valine (FVal2 performed better than FVal1, see Fig. 2 and the mass spectra in Maleckis et al., 2022).

2. **Differences in 19F chemical shifts between variants:**

   Could you briefly discuss potential reasons why the 19F chemical shifts differ between GB1-1 and GB1-d, and between GB1-2 and GB1-d? This may become clearer later in the manuscript when you discuss how rotamers are affected by fluorine substitution, but at this earlier stage the distinction is not immediately obvious to the reader.

**Response:** We propose adding (in Section 2.2 after the discussion of the $\gamma$-gauche effect) the general sentence "As the $^{19}$F chemical shifts depend on the rotamers populated by the $CH_2F$ groups as well as their chemical environment, the $^{19}$F chemical shifts observed in GB1-d do not simply recapitulate those of GB1-1 and GB1-2, necessitating an independent resonance assignment strategy." Revisiting the chemical shift description in lines 188–190, we noted that we need to include residue 21 as an example of swapped relative shifts in GB1-d.

3. **Formatting issue (page 17, line 314):**

   There appears to be a paragraph break and an unfinished sentence at this location. Please check and correct this formatting issue.

**Response:** The missing part of the sentence is "To explore the full range of the $\gamma$-effect, DFT calculations were performed". We apologize for the oversight.

4. **Fig. 12 and DFT calculations:**

   If I understand correctly, the DFT calculations do not indicate a strictly linear dependence of the FC coupling on the chemical shift, but rather a more complex relationship. Could you comment on this in the text, clarifying that the linear fit is used primarily to illustrate the presence of a correlation?

   In addition, could you discuss the relevance of the DFT calculations performed on (2R)-1-fluoro-2-methylpropane(3-13C) to the conformations of valine residues in GB1? A brief justification of this model system would be helpful.

**Response:** The sole purpose of the straight line is to guide the eye. We'll make this clearer in the legend of Fig. 12. In addition, we'll add a sentence to stress that the pseudolinearity of the correlation strengthens when all molecular degrees of freedom are relaxed in the DFT calculations (Fig. S8).

We used the model compound for DFT calculations to avoid obscuring the results by any site-specific effects in a protein context. The DFT calculations performed for 16 FVal1 residues in the protein PpiB suggested that the $^{13}C$ $\gamma$-effect is not overwhelmed by the protein context (see the first paragraph of Section 2.9).

5. **Fig. 13:**

It might be useful to include the structures of the valine residues identified in GB1-d alongside those from GB1-1 and GB1-2, to facilitate direct comparison.

**Response:** Without measurements of the $^3J_{FC}$ couplings and $^{13}C$ $\gamma$-effect, the data of GB1-d do not distinguish between $g^S$ and $g^L$ rotamers. A unique rotamer could be assigned only for the $C^{\gamma 2}H_2$ group of residue 54 based on a large $^3J_{HF}$ coupling.

In GB1-d, the $^{13}C$ $\gamma$-effect seems to be overwhelmed by other effects arising from the switch of a $CH_3$ to a $CH_2F$ group: subtracting the $^{13}C$ chemical shifts of the valine methyl groups in the wild-type protein from those of the corresponding $CH_2F$ groups in GB1-d yields $\Delta\delta(^{13}C)$ values that vary much more (> 4 ppm) than the $\Delta\delta(^{13}C)$ values reported in Table 1 for the GB1-1 and GB1-2 samples. If we ignore this and forcefully interpret the $\Delta\delta(^{13}C)$ values in GB1-d as solely reflecting the $^{13}C$ $\gamma$-effect, the preferred rotamers obtained do not correlate at all with those detected in GB1-1 and GB1-2.

6. **Clarification of 3JFC coupling statement:**

The sentence "Assigning preferential rotamers in GB1-d is more difficult, as the diFVal residues contain no CH3 group, which makes 3JFC coupling measurements difficult" is not fully clear. Could you elaborate on how the absence of a CH3 group specifically complicates the 3JFC coupling measurements?

**Response:** The transverse $^{13}C$ and $^1H$ relaxation is significantly faster for $CH_2$ than $CH_3$ groups. In addition, fewer protons contribute to the $^1H$-NMR signals.

7. **Potential applications to protein–ligand interactions:**

   It would be valuable to expand the discussion on how these fluorinated labels might be applied to studying protein–ligand interactions. Would such interactions be detectable as perturbations in the 19F chemical shifts? Do you expect these effects to be site-specific and sensitive to local changes, or rather global, given the apparent sensitivity of the 19F shifts to the overall protein structure?

**Response:** Given the sensitivity of the $^{19}$F chemical shifts to the presence of any canonical valine residues in the protein, we expect that chemical shift perturbations would not be limited to local interactions. In the case of the protein PpiB produced with Fleu or diFLeu, we demonstrated that ligand binding generates detectable changes in $^{19}$F chemical shifts, especially for residues near the ligand binding site and competitive in size with chemical shift changes observed for fluorotryptophan (Tan et al., 2024). If valine sidechains are less flexible than leucine sidechains, however, this may render them less responsive to ligand binding. We just don't know and, in the absence of any experimental data on the effects of ligand binding on the chemical shifts of FVal or diFVal residues, prefer not to speculate.

8. **Estimating energy differences between rotamers:**

   Based on your experimental data and the DFT calculations of 3JFC couplings, would it be possible to estimate the energy differences between the various rotamers?

**Response:** We'll revisit the calculations of 1-fluoro-2-methylpropane to determine energy differences.

---

## Author Response (AR1)

Dear Dr. Bax,

Thank you for encouraging submission of a final amended version. In response to the comments made, we have made the following changes.

**Reviewer 1:**

"The broadest signals were observed for the most deeply buried residues, indicating that the peak heights are sensitive indicators of the side-chain mobilities": why should deeply buried amino acids necessarily be rigid?

Response: As spelled out in the discussion phase, our observation of 19F-NMR line width correlating with depth of burial is experimental and qualitative. At this stage, we prefer not to try and interpret the relaxation rates quantitatively in the absence of more quantitative measurements.

The expression "installed" seems a bit unfortunate. How about "incorporated"? Response: We changed to "incorporated" as suggested.

Why not write out "gamma-gauche effect" without using a Greek letter, for the convenience of data bases?

Response: As spelled out in the discussion phase, we hope that data bases accept Greek/Symbol characters.

Are di-fluorinated amino acids also commercially available?

Response: We addressed the question in the discussion phase and prefer not to modify the manuscript in response.

**Reviewer 2:**

Do they care to speculate further about why some show unique 1H chemical shifts (particularly GB1-2 valine 54), while others do not?

Response: In principle, the chemical shifts of diastereotopic protons should always be different. We now mention this in line 154. We addressed the question more extensively during the discussion phase and prefer not to modify the manuscript further.

The authors note qualitative analysis of the 3J(HA-HB) couplings via analysis of COSY cross peak intensities. Is there any evidence that the intensity of those cross peaks varies

between the WT, GB-1, GB-2, and GB1-d samples, which would suggest that fluorination is affecting the chi1 populations?

Response: The signal-to-noise ratio in the DQF-COSY spectra was insufficient to draw more quantitative conclusions on the size of the  ${}^3J(\text{HA-HB})$  couplings, especially as the spectra had been recorded without  ${}^{19}\text{F}$  decoupling. We mentioned this during the discussion phase and prefer not to modify the manuscript in response.

**Minor comments:**

- On line 314, there appears to be an incomplete sentence at the beginning of the paragraph starting with a lowercase "using".

Response: The missing part of the sentence is "To explore the full range of the  $\gamma$ -effect, DFT calculations were performed". We apologize for the oversight and amended the manuscript (now line 334).

- Perhaps change "little destabilized" to "slightly destabilized" (as an adverb seems more appropriate here than an adjective) on line 538.

Response: we made the change as suggested (line 567).

- The melting temperatures were described to be about 10 °C lower than WT. However, the data for WT was not shown in Figure S1, nor given in the cited reference (Tan et al., 2024). Could the WT melting temperature be given and the source of that information be clarified? Are the reported melting temperatures from reversible or irreversible unfolding?

Response: The correct reference is Tan et al., 2025. We corrected the reference and now state in the legend of Fig. S1 that the inflection point of the heat denaturation curve is at about 79 °C for the wild-type protein.

**Reviewer 3:**

**1. Additional species in GB1-d (Fig. 2c):**

Could you expand the discussion regarding the presence of additional species observed in Fig. 2c? For instance, why are these species not present in GB1-1 and GB1-2? You write, "Therefore, the low-intensity peaks in the 19F-NMR spectrum of GB1-d most likely originate from the species with one valine and three diFVal residues." Does this imply that the absence of one diFVal residue causes a

significant change in the chemical shifts of the other 19F-labeled valines? Additionally, could you specify which residue is missing, since the small number of low-intensity peaks suggests that not all possible combinations are present in solution?

Response: We added the sentence "As two of the valine sites are close to each other in GB1, a diFVal residue in one of the sites and a valine residue in the other explains why four of the weak 19F-NMR signals in GB1-d are well resolved." in lines 91-93. More detailed reasoning was provided in the discussion phase.

**2. Differences in 19F chemical shifts between variants:**

Could you briefly discuss potential reasons why the 19F chemical shifts differ between GB1-1 and GB1-d, and between GB1-2 and GB1-d? This may become clearer later in the manuscript when you discuss how rotamers are affected by fluorine substitution, but at this earlier stage the distinction is not immediately obvious to the reader.

Response: We propose adding (lines 104-106) the general sentence "As the 19F chemical shifts depend on the rotamers populated by the CH2F groups as well as their chemical environment, the 19F chemical shifts observed in GB1-d do not simply recapitulate those of GB1-1 and GB1-2, necessitating an independent resonance assignment strategy." Revisiting the chemical shift description in lines 205–207, we also noted that we need to include residue 21 as an example of swapped relative shifts in GB1-d (line 206).

**3. Formatting issue (page 17, line 314):**

There appears to be a paragraph break and an unfinished sentence at this location. Please check and correct this formatting issue.

Response: The missing part of the sentence is "To explore the full range of the  $\gamma$ -effect, DFT calculations were performed" (line 334). We apologize for the oversight.

**Fig. 12 and DFT calculations:**

If I understand correctly, the DFT calculations do not indicate a strictly linear dependence of the FC coupling on the chemical shift, but rather a more complex relationship. Could you comment on this in the text, clarifying that the linear fit is used primarily to illustrate the presence of a correlation?

Response: The sole purpose of the straight line is to guide the eye. We have made this clearer in the legend of Fig. 12. In addition, we added a sentence in lines 364-366 to stress that the pseudolinearity of the correlation is better when all molecular degrees of freedom are relaxed in the DFT calculations (Fig. S8c).

In addition, could you discuss the relevance of the DFT calculations performed on (2R)-1-fluoro-2-methylpropane(3-13C) to the conformations of valine residues in GB1? A brief justification of this model system would be helpful.

Response: The justification for the model compound is given in line 334: we used it for the DFT calculations to avoid obscuring the results by site-specific effects in a protein context.

**4. Fig. 13:**

It might be useful to include the structures of the valine residues identified in GB1-d alongside those from GB1-1 and GB1-2, to facilitate direct comparison.

Response: As explained during the discussion phase, without measurements of the  ${}^3J_{FC}$  couplings and  ${}^{13}C$   $\gamma$ -effect, the data of GB1-d do not distinguish between  $\gamma^S$  and  $\gamma^L$  rotamers. A unique rotamer could be assigned only for the  $C^{\gamma 2}H_2$  group of residue 54 based on a large  ${}^3J_{HF}$  coupling. For the sake of conciseness, we prefer not to reiterate in detail, why we cannot assign more rotamers in GB1-d.

**5. Clarification of 3JFC coupling statement:**

The sentence "Assigning preferential rotamers in GB1-d is more difficult, as the diFVal residues contain no CH3 group, which makes 3JFC coupling measurements difficult" is not fully clear. Could you elaborate on how the absence of a CH3 group specifically complicates the 3JFC coupling measurements?

Response: As mentioned during the discussion phase, the transverse 13C and 1H relaxation is significantly faster for CH2 than CH3 groups. In addition, fewer protons contribute to the 1H-NMR signals. We added the words "slowly relaxing" in line 416.

**6. Potential applications to protein-ligand interactions:**

It would be valuable to expand the discussion on how these fluorinated labels might be applied to studying protein–ligand interactions. Would such interactions

be detectable as perturbations in the 19F chemical shifts? Do you expect these effects to be site-specific and sensitive to local changes, or rather global, given the apparent sensitivity of the 19F shifts to the overall protein structure?

Response: As mentioned in the discussion phase, we expored the sensitivity of the 19F chemical shifts in the protein PpiB produced with Fleu or diFLeu (Tan et al., 2024). As we don't have a ligand that binds to GB1, we prefer not to speculate.

**7. Estimating energy differences between rotamers:**

Based on your experimental data and the DFT calculations of 3JFC couplings, would it be possible to estimate the energy differences between the various rotamers?

Response: We now present the calculated energy differences for 1-fluoro-2-methylpropane in Figure S8d.

Finally, we came across early references by Tonelli et al. describing the  $\gamma$ -effect on  $^{13}$ C chemical shifts in fluorinated polymers. We amended the statement in lines 521-522 accordingly and added two references. In addition, we took out the hyphen from "sidechain", which also changed the title.

Thank you for guiding the editorial process.